# Effects of Virtual Reality-Based Cognitive Rehabilitation in Stroke Patients: A Randomized Controlled Trial

**DOI:** 10.3390/healthcare11212846

**Published:** 2023-10-29

**Authors:** Mingyeong Park, Yeongmi Ha

**Affiliations:** 1Depart of Nursing, Jinju Health College, Jinju 52655, Republic of Korea; hongeai@naver.com; 2College of Nursing & Institute of Medical Science, Gyeongsang National University, Jinju 52727, Republic of Korea

**Keywords:** stroke, virtual reality, cognitive training, self-efficacy

## Abstract

During the process of recovering functional ability after damage caused by a stroke, it is important to restore cognitive function via cognitive rehabilitation. To achieve successful rehabilitation, it is important for patients to have a sense of efficacy in their ability to manage their disease well. Therefore, a virtual reality-based cognitive rehabilitation program based on self-efficacy theory was developed, and its effects were compared with conventional and computer-assisted cognitive rehabilitation. The virtual reality-based cognitive rehabilitation program consisted of sessions lasting 30 min each five days a week for eight weeks. After applying the virtual reality-based cognitive rehabilitation program, there were significant differences in group-by-time interactions regarding stroke self-efficacy, cognitive function, visual perception, activities of daily living, and health-related quality of life. In addition, there were significant group differences among the three groups in terms of stroke self-efficacy and health-related quality of life. In conclusion, our virtual reality-based cognitive rehabilitation program developed based on self-efficacy theory is effective for inpatients with stroke and improves their stroke self-efficacy, cognitive function, visual perception, activities of daily living, and health-related quality of life.

## 1. Introduction

Stroke is a chronic neurological condition caused by cerebrovascular diseases, which can lead to a number of dysfunctions. According to Statistics Korea, stroke was the second most common cause of death in 2018 and the fourth most common cause in 2020 [1]. Although the mortality rate associated with stroke is decreasing, longer life expectancy is contributing to an increase in its prevalence [2]. Advances in medical technology have made it possible to deal with stroke in its early stages, leading to a lower mortality rate [3]. However, the number of stroke survivors with various complications is continuously rising [3].

According to the Clinical Practice Guidelines for Stroke Rehabilitation [4], the success of rehabilitation is influenced by the degree of cognitive function impairment. Stroke patients with cognitive impairment struggle to carry out tasks, solve problems, and process information due to pervasive afflictions associated with attention, memory, comprehension, and problem-solving abilities [5]. Over 40% of stroke patients have cognitive impairment, and two-thirds are at risk of dementia. This is why cognitive rehabilitation for stroke patients is crucial [6]. The significance of cognitive rehabilitation has been increasingly recognized ever since cognitive rehabilitation for stroke survivors was found to effectively improve their cognitive function and ability to perform daily activities [7,8,9].

Therapeutic interventions for cognitive rehabilitation include conventional cognitive rehabilitation, computer-assisted cognitive rehabilitation (CACR), and virtual reality (VR)-based cognitive rehabilitation. Conventional cognitive rehabilitation typically involves an occupational therapist working one-on-one with patients to improve their attention, memory, visual perception, and problem-solving abilities [10,11]. CACR uses a computer system for cognitive rehabilitation, which is similar to conventional cognitive rehabilitation but tailored to each patient’s level [12]. VR-based cognitive rehabilitation provides visuospatial stimuli similar to real-life experience while possessing the distinct features of immersion, interaction, and realism [5]. Conventional training or CACR comprises independent activity-oriented programs to enhance specific components of cognitive functions, such as attention, memory, and visual perception [13,14]. According to previous systematic reviews and studies on the use of computer technologies for cognitive rehabilitation, such technologies can improve cognitive function and problem-solving abilities in people with stroke, traumatic brain injury, Alzheimer’s disease, and Parkinson’s disease [15,16,17,18]. Despite the positive outcomes of conventional training or CACR in improving cognitive function, these cognitive rehabilitation programs have limitations in that individual components of cognitive functions are trained separately even though stroke patients require a combination of attention, memory, and visual perception in their actual daily living environment [4,13,14]. VR-based cognitive rehabilitation can effectively merge separate cognitive rehabilitation domains by artificially generating visual, auditory, and tactile stimuli to create complex and integrated everyday situations [19]. Furthermore, previous studies have shown promising results in using virtual reality-based cognitive rehabilitation to improve the cognitive function of patients with stroke or traumatic brain injury [6,20,21].

Self-efficacy can influence how effectively people manage chronic diseases by motivating health-promoting behaviors [22]. The motivational outcomes of self-efficacy include choice of activities, effort, persistence, and achievement [22]. Patients with stroke need to develop a sense of efficacy to complete long-term rehabilitation [23]. Self-efficacy in stroke patients helps them achieve their goals regarding successful illness management and influences their motivation for rehabilitation, thus ultimately having a positive impact on their quality of life [19,23,24]. Higher self-efficacy can be constructed from four principal sources, namely performance accomplishment, vicarious experiences, verbal persuasion, and physiological/affective states [22]. Enhancing the self-efficacy of stroke patients is crucial for successful cognitive rehabilitation [23]. Therefore, this study aimed to develop a VR-based cognitive rehabilitation program based on self-efficacy theory and verify the effects of cognitive rehabilitation on stroke self-efficacy, cognitive function, visual perception, activities of daily living (ADL), and health-related quality of life (HRQoL).

## 2. Materials and Methods

### 2.1. Study Design

This study is a three-armed randomized controlled trial with a pretest–posttest equivalent control group to examine the effects of three different cognitive rehabilitation programs in patients with stroke.

### 2.2. Participants

The target population was inpatients diagnosed with stroke and receiving cognitive rehabilitation. The specific inclusion criteria were as follows: patients diagnosed with stroke for the first time; patients hospitalized and undergoing cognitive rehabilitation as a result of a stroke; patients whose onset of stroke was 3 to 36 months ago; patients who scored between 17 and 27 on the Korean version of the Mini-Mental State Examination, Second Edition (K-MMSE-2); and patients who could participate in the survey with the help of a guardian. The specific exclusion criteria were as follows: patients who scored between 0 and 24 on the Korean version of the Modified Barthel Index (K-MBI); patients with visuospatial neglect; patients with underlying neurological conditions, such as dementia; and patients with a history of any psychiatric disorders due to medical concerns, in whom immersive VR might cause or exacerbate certain psychosocial symptoms as side effects, such as a reduced sense of presence and the development of inadequate responses to the real world [25].

The sample size was determined using the G*power 3.1.9.7 program. Considering the three groups and three measurement points, a total of 45 subjects would be required to achieve a medium effect size of 0.25, a significance level of 0.05, a power of 0.90, and a correlation coefficient of 0.5 for repeated measurement. Given the dropout rate, a total of 64 participants were recruited.

A total of 64 participants were enrolled and hospitalized in 18 patient rooms, with 3 to 4 inpatients housed in each 4-bed room. Of the 64 participants, one individual who did not meet the inclusion criteria was excluded. The other participants were grouped by patient room, which was designated to one of the 18 blocks, to prevent the likelihood of treatment contamination. Participants in each room were randomly assigned to one of the three groups using a computerized random code generator. A total of 21 participants were assigned to the experimental group, 22 participants were assigned to control group 1, and 20 participants were assigned to control group 2. One individual from the experimental group later dropped out due to personal reasons, and two individuals from control group 1 dropped out due to personal reasons and hospital discharge. At the end, 60 participants were included in the analysis (Figure 1).

### 2.3. Measurements

Demographic characteristics, including gender, age, educational level, job status, marital status, and subjective socioeconomic status, were measured in all participants. Disease-related characteristics, including subtypes of stroke, operation history, and underlying diseases, were also assessed.

#### 2.3.1. Stroke Self-Efficacy

The stroke self-efficacy questionnaire developed by Jones et al. [26] was used to measure self-efficacy judgments in specific domains of cognitive functioning among the participants. This scale encompasses tasks related to various activities (such as mobility, eating, walking, and dressing) and self-management (such as coping, overcoming obstacles, and exercise). It consists of 13 items rated on an 11-point ordinal scale ranging from 0 points, indicating ‘not at all confident’, to 10 points, signifying ‘very confident’. The range of possible scores is 0–130, with higher scores indicating higher self-efficacy. The reliability of this scale was high (0.90) in this study.

#### 2.3.2. Cognitive Function

The Korean version of the Mini-Mental State Examination, 2nd edition, specifically the standard version (K-MMSE~2:SV) of the Korean Dementia Association [27], was used to assess cognitive impairment. This tool is composed of six subtests: recall, attention, registration, orientation, calculation, language, and visual-constructional ability. The 30 items of this test are scored on a scale ranging from 0 to 30, with higher scores indicating better cognitive function. The reliability of the K-MMSE~2 was high (0.95) in this study.

#### 2.3.3. Visual Perception

The Motor-Free Visual Perception Test, 3rd edition (MVPT-3), developed by Colarusso and Hammill [28], was used to assess visual perception functioning in adults. The MVPT-3 consists of 5 visual domains: visual discrimination, visual figure–ground discrimination, visual memory, visual closure, and visual–spatial perception. Each item consists of a black-and-white line-drawing stimulus and four multiple-choice response options for participants to choose the option that matches the stimulus. The total score ranges from 13 to 65. Higher scores indicate fewer deficits in general visual perceptual function. The reliability of the MVPT-3 was high (0.83) in this study.

#### 2.3.4. Activities of Daily Living (ADL)

The Korean version of the Modified Barthel Index (K-MBI) was used to evaluate participants’ capabilities in daily life activities [29]. The K-MBI is composed of 10 items: feeding, bathing, grooming, toilet use, stair climbing, dressing, bowel control, bladder control, mobility on level surfaces, and chair/bed transfers. The K-MBI has three different rating scales with a score range of 0–5 (bathing and grooming), a score range of 0–10 (feeding, dressing, toilet use, bladder control, bowel control, and stair climbing), and a score range of 0–15 (chair/bed transfers and walking). It is scored on a scale from 0 to 100, with higher scores representing a higher degree of independence in performing activities of daily living. The reliability of the K-MBI was high (0.88) in this study.

#### 2.3.5. Health-Related Quality of Life (HRQoL)

After obtaining permission to use the SF-12 from QualityMetric, the 12-Item Short Form Health Survey (SF-12) developed for the Medical Outcomes Study [30] was used to assess health-related quality of life. The SF-12 comprises two components, including a physical health component and a mental health component. It includes eight health-related domains: general health, physical functioning, role limitations due to physical health, pain, vitality, social functioning, role limitations due to emotional health, and mental health. It is scored on a scale ranging from 0 to 100 points, with higher scores representing a higher degree of quality of life in terms of health-related aspects. The reliability of the SF-12 was high (0.85) in this study.

### 2.4. Interventions

#### 2.4.1. Development of VR-Based Cognitive Rehabilitation Based on Self-Efficacy Theory

The VR-based cognitive rehabilitation program grounded in self-efficacy theory was offered five days a week for eight weeks, with a total of 40 sessions. Each session lasted 30 min (15 min of virtual reality-based cognitive rehabilitation and 15 min of training tasks related to VR contents using a stroke cognitive recovery workbook). For the VR-based cognitive rehabilitation program to improve the cognitive function of the experimental group, virtual reality contents (InTheTech Co., Ltd., Daegu, Republic of Korea), named ‘Popping colorful balloons’, ‘Finding same fishes’, ‘Save the planet’, ‘Throwing dart’ and ‘Other contents,’ were modified with consideration of each stroke patient’s level of cognitive function, and the ‘Stretching and aerobic exercise in the park’ and ‘Choosing best foods for stroke patients’ contents were added to customize the educational needs of the stroke patients in this study. For example, the ‘Finding same fishes’ content involved locating and eliminating a target fish while avoiding other fish, and ‘Popping colorful balloons’ required removing balloons that appeared on the screen in a specific order based on their shape and color. The ‘Save the planet’ content involved avoiding obstacles while finding and rescuing planets that matched the ones displayed on both sides of the screen. Especially in ‘Stretching and aerobic exercise in the park’, participants performed several missions related to exercise while walking in the park. These missions included activities such as picking up leaves, avoiding bicycles, and giving a high five to dogs. The contents were classified into levels 1 through 5. Each of the contents was designed to train a specific cognitive function, such as attention, memory, visual perception, and executive function. Individualized training sessions were conducted, and the activities varied daily to align with the cognitive domains trained in the virtual reality contents. For example, activities such as rhythmic clapping, cup tapping, singing with modified lyrics, and learning rhythmic patterns were performed. By the end of a daily VR activity, homework related to the VR contents was given to the experimental group every day. Additionally, participants in the experimental group had small-group discussions of four or five persons to share their VR experiences every Friday. To encourage active participation in cognitive rehabilitation, participants were given strategies to enhance performance accomplishments (adjusting the difficulty of virtual reality contents), vicarious experience (small-group discussions on success stories), and verbal persuasion (support, encouragement, and constructive feedback).

#### 2.4.2. Experimental Group

The patient-specific individualized program was provided five days a week for eight weeks from June to August 2022. The experimental group participated in 30 min of conventional cognitive rehabilitation in the morning and 30 min of VR-based cognitive rehabilitation in the afternoon. A therapist engaged with the participants one-on-one and conducted conventional cognitive rehabilitation using papers and pencils for cognitive activities, such as cards, puzzles, calculation tasks, and picture and number matching. VR-based cognitive rehabilitation based on self-efficacy theory was given to the participants in a virtual reality room. The VR-based cognitive rehabilitation was conducted using immersive programs with goggles and controllers, with each session lasting 30 min and consisting of 15 min of virtual reality-based cognitive rehabilitation and 15 min of individual training using a stroke cognitive recovery workbook. Furthermore, group discussions were conducted with four or five participants in each group every Friday.

#### 2.4.3. Control Groups

Control group 1 received 30 min of conventional cognitive rehabilitation in the morning and 30 min of computer-assisted cognitive rehabilitation in the afternoon five days a week for eight weeks. Computer-assisted cognitive rehabilitation was administered by a therapist in the rehabilitation therapy room using ComCog (Neofect, Seongnam, Republic of Korea, 2019), where patients interacted with a computer monitor under the guidance of the therapist. The contents of ComCog consist of attention and memory training exercises, including activities such as puzzle solving and piano playing, spanning 16 levels in total. ComCog was designed to be immersive and enjoyable and provided quantitative feedback on patients’ activities. The difficulty level of training tasks could be continually adjusted based on each patient’s performance.

Control group 2 received 30 min of conventional cognitive rehabilitation each morning and afternoon administered by an occupational therapist five days a week for eight weeks in the rehabilitation therapy room. The conventional cognitive rehabilitation administered by the therapist involved a structured approach to improve participants’ cognitive functions. The training sessions were conducted one-one-one using paper-and-pencil tasks, which consisted of puzzles, calculation, picture matching, and others.

### 2.5. Data Collection

This study consisted of preliminary, interim, and final examinations. One occupational therapist and one survey conductor were hired to collect data. An occupational therapist with at least five years of experience conducting cognitive rehabilitation was hired to assess cognitive function, visual perception, and ADL tests. A nurse with at least five years of clinical nursing experience was hired to conduct the survey. Neither the occupational therapist nor the survey conductor knew which participants were assigned to the experimental or control groups. The researchers held an hour-long training session with the survey conductor on how to use the survey instruments and had the survey conductor run a preliminary survey round to minimize measurement errors.

The pretest was conducted in June 2022, when the survey conductor personally visited the participants’ hospital rooms. Individual responses to the survey were taken when a guardian or caregiver was present. The occupational therapist administered cognitive function, visual perception, and ADL tests in the treatment room. The interim examination was conducted by the same survey conductor and occupational therapist four weeks after the beginning of the program, and the posttest was conducted shortly after the end of the eight-week program.

### 2.6. Data Analysis

The collected data were analyzed using the IBM SPSS/WIN version 27.0 program. First, ANOVA and a chi-squared test with Fisher’s exact test were used to analyze the homogeneity of the participants. The Shapiro–Wilk test was performed to examine the normality of the dependent variables. Second, repeated-measures ANOVA was used to verify the effects of the study variables. Before running the analysis, the assumptions of normality, homogeneity, and sphericity of the dependent variables required for repeated-measures ANOVA were confirmed. Partial eta squared (η_p_^2^) was used to estimate the effect size to more clearly understand the comparability across between-subject and within-subject designs. The Bonferroni method was used as a post hoc test for the ANOVA; this method is commonly used to assess the significance of differences among three or more groups.

## 3. Results

### 3.1. Homogeneity Test of Demographic and Disease-Related Characteristics of Participants

There were no significant differences in demographic and disease-related characteristics among the experimental group, control group 1, and control group 2. Almost half of the participants were male, and their mean age was 62.5 years. The educational level of the participants ranged from elementary school to college, and the proportion was almost evenly distributed across the educational levels. Most participants responded that their subjective economic status was moderate. Over half of the participants had a diagnosis of ischemic stroke, and 13.3% of the participants had surgery. Most participants had underlying diseases. At pretest, there were no significant differences among the three groups in terms of stroke self-efficacy, cognitive function, visual perception, ADLs, and HRQO (Table 1).

### 3.2. Effects of VR-Based Cognitive Rehabilitation Developed Based on Self-Efficacy Theory

The results regarding the effects of VR-based cognitive rehabilitation grounded in self-efficacy theory are presented in Table 2. There were significant group differences among the three groups in terms of stroke self-efficacy (F = 6.61, *p* = 0.003, η_p_^2^ = 0.10) and HRQoL (F = 5.09, *p* = 0.009, η_p_^2^ = 0.11).

After applying the intervention, there were significant differences at different time points in terms of stroke self-efficacy (F = 71.16, *p* < 0.001), cognitive function (F = 65.28, *p* < 0.001), visual perception (F = 340.13, *p* < 0.001), ADL (F = 48.86, *p* < 0.001), and HRQoL (F = 342.28, *p* < 0.001).

In the interaction between groups and time points, there were statistically significant differences in terms of stroke self-efficacy (F = 78.62, *p* < 0.001), cognitive function (F = 9.33, *p* < 0.001), visual perception (F = 14.66, *p* < 0.001), ADLs (F = 14.15, *p* < 0.001), and HRQoL (F = 213.87, *p* < 0.001).

## 4. Discussion

The current study implemented a VR-based cognitive rehabilitation program developed based on self-efficacy theory for stroke patients, and its effectiveness was compared to that of computer-assisted and conventional cognitive rehabilitation. It is advisable to start cognitive and physical rehabilitation within 3 to 36 months after the onset of stroke [31]. This study is meaningful because it employed a VR-based cognitive rehabilitation program developed based on self-efficacy theory for stroke patients during this critical period and improved their self-efficacy in managing their condition, thus further enhancing their cognitive function, visual perception, ADLs, and HRQoL.

Eight weeks into the VR-based cognitive rehabilitation program, the stroke self-efficacy of the experimental group showed significant changes at various points of measurement, while no significant differences in self-efficacy were observed over time in control group 1 and control group 2. By the fourth week of the intervention, the experimental group’s self-efficacy improved; by week 8, it had significantly increased. This is consistent with a previous study that compared the effectiveness of programs using self-efficacy improvement strategies and traditional rehabilitation programs in stroke patients [32]. Various strategies for improving self-efficacy, such as performance accomplishments, vicarious experience, and verbal persuasion, contributed to the higher self-efficacy of stroke patients in the experimental group compared to the control groups at weeks 4 and 8. The participants in the experimental group were aided in achieving success through virtual reality contents catered to each participant’s level, workbook training, and task performance. In addition, sharing their experiences related to stroke management in weekly face-to-face group discussions among stroke patients helped these participants feel more confident about self-management of strokes. Self-efficacy plays a vital role in stroke patients’ ability to self-manage their condition throughout their lives to prevent stroke recurrence and overcome challenges [20,23,33]. Therefore, it is particularly notable that patients who underwent VR-based cognitive rehabilitation in this study experienced a significant increase in self-efficacy in managing this chronic disease.

As expected, the cognitive function of the experimental group, control group 1, and control group 2 was significantly different at each measurement point. The cognitive function of the experimental group was higher than that of control group 1 and control group 2 in the eighth week, but there were not significant group differences among the three groups. According to a previous study on the effects of VR-based cognitive rehabilitation and conventional cognitive rehabilitation in stroke patients, both the experimental and control groups showed elevated cognitive function, though more so in the experimental group [6,20]. We believe the experimental group had higher cognitive function at week 8 due to the use of virtual reality contents that combined various cognitive function-related training modules rather than due to independent activity-oriented cognitive rehabilitation. While conventional cognitive rehabilitation has the advantage of providing cognitive training on attention, execution, visuospatial perception, and memory as segmental contents, VR-based cognitive rehabilitation can effectively integrate these area-specific activities and train patients in combination [14]. For instance, it is believed that the virtual reality content “Choosing best foods for stroke patients” had an impact on the cognitive function of the experimental group because it was created to naturally employ various cognitive dimensions, such as selecting menus, making shopping lists, successfully navigating obstacles on the way to a grocery store, and calculating prices of foods. Since the improvement in stroke patients’ cognitive function is considered a crucial indicator of the success of rehabilitation, it is necessary to pay attention to the significant improvement in cognitive function due to the experimental intervention in this study; this finding supports actively utilizing it in nursing practice.

At each measurement point, the three groups showed a significant improvement in visual perception, with the visual perception of the experimental group at week 8 being higher than that of control groups 1 and 2, but a between-group analysis reported that there was no significant difference. According to a prior study comparing the effectiveness of VR-based cognitive rehabilitation and computer-assisted cognitive rehabilitation in stroke patients, the visual perception of both the experimental and control groups improved, with a significant improvement in the experimental group [33]. The use of virtual reality content to improve visual perception resulted in superior visual perception in the experimental group compared to the control groups at week 8 of the intervention. For example, in the “Save the Planet” virtual reality content, participants selected and matched planets of identical shape, separated and classified different planetary shapes using each hand, and searched for a specific planet among other similar planets. Visual perception, which requires the brain to interpret visual stimuli, is the ability to process and understand visual stimuli to perceive what is seen. The experimental group in this study experienced spatiotemporal stimuli similar to real-life experience via virtual reality, which also helped improve these participants’ visual perception.

One of the interesting findings is that ADLs were significantly different at each measurement point among all three groups, and the ADLs of the experimental group were rated higher than that of control groups 1 and 2 at week 8, but there was no significant difference in a between-group analysis. A previous study comparing the effects of VR-based cognitive rehabilitation and conventional cognitive rehabilitation in stroke patients demonstrated higher ADLs in both groups, with the experimental group reporting a greater increase [34]. The higher ADLs in the experimental group compared to the control groups at week 8 likely resulted from the regular use of virtual reality content related to physical activity to improve ADLs every week. The virtual reality experience “ Stretching and aerobic exercise in the park” allowed the participants to train the muscles in their upper and lower extremities every week. In a systematic review of their effectiveness for stroke patients, VR programs were found to be a safe intervention to improve ADLs in stroke patients; however, the results varied from study to study [4]. More studies on the effects of VR-based cognitive rehabilitation on ADLs in stroke patients are needed.

The HRQoL of all three groups varied significantly at each measurement point in this study. The HRQoL of the experimental group increased until the fourth week of the intervention and then rose significantly at week 8. These findings differ from those of previous studies on the effect of VR programs on stroke patients’ quality of life, wherein they were found to be effective in one study [35] and weak or ineffective in some other studies [4,36]. In this study, the HRQoL of the experimental group was higher than that of the control groups at week 8 of the experimental intervention because these participants could participate more intensively in the program as their self-efficacy increased. Thus, it can be deduced that these participants’ improved cognitive function, visual perception, and ADLs positively affected their overall HRQoL. It is crucial to focus on the active participation of stroke patients in cognitive rehabilitation to improve their cognitive function and quality of life. This is because the sequelae of stroke may cause lower ADLs, visual perception impairment, and cognitive impairment, as well as feelings of helplessness and depression, when this condition lasts for a long time, thus potentially impacting patients’ quality of life in turn [26,37].

This study is significant as it is the first to develop an immersive VR-based cognitive rehabilitation program using Bandura’s self-efficacy improvement strategies and compare its effectiveness with conventional and computer-assisted cognitive rehabilitation. Notably, while therapists have previously led cognitive rehabilitation for stroke patients, this program facilitates the self-management of stroke patients by presenting a nurse-led VR-based cognitive rehabilitation tailored to each individual. Despite its innovative significance, our study has some limitations. First, there are limitations with regard to generalizing the results to all stroke patients with cognitive impairment because this study was conducted with patients admitted to a rehabilitation hospital in Korea. Second, this study was conducted with patients who experienced a stroke between 3 and 36 months ago, which may limit the generalizability of the research findings. Third, while a partial eta squared value indicating a small effect size might be statistically significant, it might not be meaningful in terms of tangible clinical outcomes in the real world. That means that there is a need to pay attention to the interpretation of research findings because partial eta squared values might potentially overestimate the true effect size in smaller randomized controlled trials.

## 5. Conclusions

Using a three-arm randomized controlled trial design, this study built a VR-based cognitive rehabilitation program based on self-efficacy theory by applying various self-efficacy strategies, such as performance accomplishments, vicarious experience, and verbal persuasion, and verified its effects. Our findings demonstrated that the HRQoL and self-efficacy of stroke patients in the experimental group improved significantly more than those in control group 1, who received conventional rehabilitation, and control group 2, who underwent computer-assisted cognitive rehabilitation. In addition, there were significant differences in group-by-time interactions regarding stroke self-efficacy, cognitive function, visual perception, activities of daily living, and HRQoL.

The following recommendations are made based on the results of the current study. First, it is proposed that further studies be conducted to validate the effectiveness of the immersive VR-based cognitive rehabilitation program developed based on self-efficacy theory among stroke patients who are admitted to various types of hospitals, such as tertiary general hospitals and other general hospitals. Second, a previous study reported that the cognitive function of stroke patients significantly improved up to 12 months after a stroke, but the improvement was only marginal after 36 months [31]. Therefore, it is necessary to investigate the program’s effect according to the onset date, such as between 3 and 12 months or between 12 and 36 months.

## Figures and Tables

**Figure 1 healthcare-11-02846-f001:**
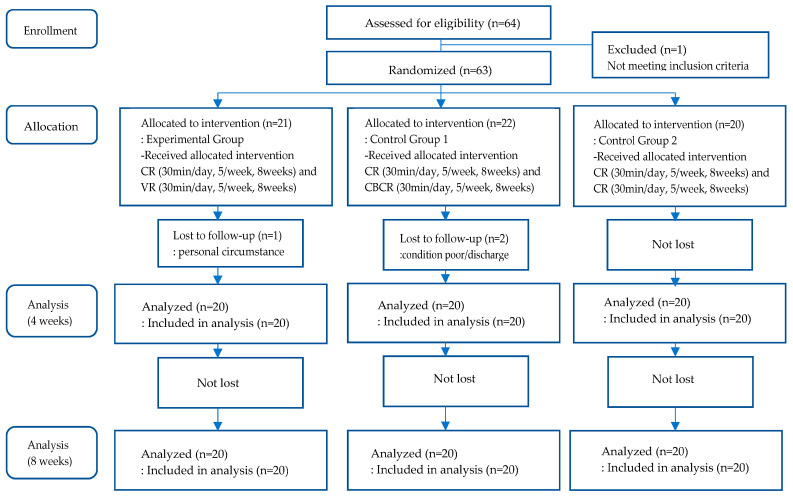
Flow diagram of VR-based cognitive rehabilitation for stroke patients.

**Table 1 healthcare-11-02846-t001:** Homogeneity test of demographic and disease-related characteristics of the participants.

Characteristics	Categories	Exp (*n* = 20) *n* (%)	Cont 1 (*n* = 20) *n* (%)	Cont 2 (*n* = 20) *n* (%)	X^2^ or F (*p*)
Gender	Male	11 (55.0)	12 (60.0)	12 (60.0)	0.06 (0.937)
	Female	9 (40.0)	8 (40.0)	8 (40.0)	
Age(year)		62.5 ± 4.7	62.0 ± 3.0	62.5 ± 4.8	0.07 (0.929)
Educational level	Elementary school	4 (20.0)	5 (25.0)	5 (25.0)	0.59 (0.558)
	Middle school	6 (30.0)	7 (35.0)	5 (25.0)	
	High school	6 (30.0)	6 (30.0)	4 (20.0)	
	College	4 (20.0)	2 (10.0)	6 (30.0)	
Job status	Yes	7 (35.0)	6 (30.0)	5 (25.0)	0.71 (0.494)
	No	13 (65.0)	14 (70.0)	15 (75.0)	
Marital status	Married	19 (95.0)	18 (90.0)	19 (95.0)	0.96 (0.388)
	Unmarried	1 (5.0)	2 (10.0)	1 (5.0)	
Subjective SES	High	0 (0.0)	0 (0.0)	3 (15.0)	0.23 (0.794)
	Moderate	20 (100.0)	19 (95.0)	14 (70.0)	
	Low	0 (0.0)	1 (5.0)	3 (15.0)	
Subtype of stroke	Hemorrhagic stroke	9 (45.0)	10 (50.0)	8 (40.0)	0.19 (0.825)
	Ischemic stroke	11 (55.0)	10 (50.0)	12 (60.0)	
Operation history	Yes	3 (15.0)	3 (15.0)	2 (10.0)	0.13 (0.872)
	No	17 (85.0)	17 (85.0)	18 (90.0)	
Underlying disease	Yes	17 (85.0)	13 (65.0)	17 (85.0)	0.57 (0.216)
	No	3 (15.0)	7 (35.0)	3 (15.0)	
Stroke self-efficacy		61.40 ± 6.37	61.60 ± 13.70	56.15 ± 20.71	0.64 (0.527)
Cognitive function		20.35 ± 1.56	20.20 ± 1.82	19.85 ± 1.63	0.46 (0.629)
Visual perception		37.10 ± 2.57	37.00 ± 2.82	37.25 ± 1.94	0.52 (0.950)
ADL		56.80 ± 12.78	56.20 ± 8.91	55.05 ± 7.78	0.15 (0.856)
HRQoL		48.43 ± 4.86	48.87 ± 3.53	47.78 ± 8.01	0.18 (0.836)

Exp = experimental group; Cont 1 = control group 1; Cont 2 = control group 2; SES = socioeconomic status; ADL = activities of daily living; HRQoL = health-related quality of life.

**Table 2 healthcare-11-02846-t002:** Effects of VR-based cognitive rehabilitation developed based on self-efficacy theory.

Variable	Group	Pretest M ± SD	4 Weeks M ± SD	8 Weeks M ± SD	Effect by Time Point F (*p*)	F *(p)*		Effect Size (η_p_^2^) [95% CI]
Stroke self-efficacy	Exp	61.40 ± 6.37	70.40 ± 14.11	89.00 ± 8.14	107.92 (<0.001)	Group	6.61 (0.003)	0.10 [0.03–1.00]
	Cont 1	61.60 ± 13.70	61.30 ± 13.52	62.05 ± 13.84	3.18 (0.053)	Time	71.16 (<0.001)	
	Cont 2	56.15 ± 20.71	55.80 ± 20.59	54.75 ± 21.11	0.72 (0.490)	GroupxTime	78.62 (<0.001)	
Cognitive function	Exp	20.35 ± 1.56	20.45 ± 1.57	21.70 ± 1.62	49.42 (<0.001)	Group	1.26 (0.290)	0.00 [0.00–1.00]
	Cont 1	20.20 ± 1.82	20.40 ± 1.84	21.05 ± 1.79	18.53 (<0.001)	Time	65.28 (<0.001)	
	Cont 2	19.85 ± 1.63	20.00 ± 1.62	20.20 ± 1.67	4.91 (0.030)	GroupxTime	9.33 (<0.001)	
Visual perception	Exp	37.10 ± 2.57	39.30 ± 2.65	43.10 ± 3.30	111.49 (<0.001)	Group	1.30 (0.280)	0.03 [0.00–1.00]
	Cont 1	37.00 ± 2.82	37.10 ± 3.04	43.05 ± 3.63	207.46 (<0.001)	Time	340.13 (<0.001)	
	Cont 2	37.25 ± 1.94	38.45 ± 2.01	40.20 ± 2.11	53.43 (<0.001)	GroupxTime	14.66 (<0.001)	
ADL	Exp	56.80 ± 12.78	59.35 ± 13.16	65.85 ± 12.7	128.99 (<0.001)	Group	1.50 (0.230)	0.02 [0.00–1.00]
	Cont 1	56.20 ± 8.91	60.15 ± 9.20	61.10 ± 9.12	7.31 (0.013)	Time	48.86 (<0.001)	
	Cont 2	55.05 ± 7.78	56.25 ± 8.10	56.75 ± 8.09	7.33 (0.002)	GroupxTime	14.15 (<0.001)	
HRQoL	Exp	48.43 ± 4.86	50.56 ± 5.40	56.87 ± 5.65	130.51 (<0.001)	Group	5.09 (0.009)	0.11 [0.03–1.00]
	Cont 1	48.87 ± 3.53	49.30 ± 3.75	49.91 ± 3.69	23.47 (<0.001)	Time	342.28 (<0.001)	
	Cont 2	47.78 ± 8.01	48.37 ± 7.79	48.65 ± 8.00	20.07 (<0.001)	GroupxTime	213.87 (<0.001)	

M = mean; SD = standard deviation; CI = confidence interval; Exp = experimental group; Cont 1 = control group 1; Cont 2 = control group 2; ADL = activities of daily living; HRQoL = health-related quality of life.

## Data Availability

The datasets generated and/or analyzed during the current study are not publicly available due to concerns regarding patient privacy. The data presented in the study can be made available from the corresponding author upon reasonable request.

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
