# Peer review of "Effects of Virtual Reality-Based Cognitive Rehabilitation in Stroke Patients: A Randomized Controlled Trial"

_healthcare, 2023, doi:10.3390/healthcare11212846_

Round 1
Reviewer 1 Report
Comments and Suggestions for Authors
This is an interesting study investigating a new VR-based cognitive rehabilitation approach in addition to conventional cognitive rehabilitation and compared to a conventional plus computer-based (but not VR) cognitive rehabilitation as control group 1 and a dose-and-schedule matched conventional control group 2. The following are strongly recommended to strengthen the manuscript.
1). Introduction - last paragraph - It is not clear what is meant by a cognitive rehabilitation that is based on self-efficacy theory and a theory can't "manage" anything. Please make this paragraph much clearer so the reader has a good idea of what is meant.
2) Methods - there needs to be a better description of the VR-based rehabilitation. There was some more description in the discussion, but it would be better put here so that the reader really understands the difference between the interventions. Also, since computer -based rehab has sometimes been considered VR, what is the difference between the computer-based cognitive rehab and the VR-based cognitive rehab.
3) Results - Table 2 is confusing - the means columns are clear, but the way the analyses columns are provided is confusing, having a time column but also a time row and then the G x T only within 1 row, etc.
4) Discussion - while there is a statistical difference favoring the VR group for the outcomes, often the differences between groups is quite small. This raises the issue of whether or not these are meaningful differences. Do they meet or exceed the MCID or SEM for these measures?
Small typo edits
1) lines 105 and table vs line 96 - these give two different numbers for the number of enrollees. Please reconcile.
2) line 56 - please reword "individual independent cognitive rehab. I think you mean cognitive training where individual components of cognition are trained separately?
3) Consort diagram - you have 2 control group I's
4) lines 198-199 vs 205 - did this group get 2 30-min sessions/day or was one really 15 min/day?
5) line 298 - so there was a regular group discussion as part of the VR group's rehab? If so this should be in the methods. If not, then it needs to be explained how this occurred if it wasn't part of the study and could be a confound.
Author Response
- Summary
We sincerely thank the reviewer's valuable comments for thorough reading of our manuscript. Our manuscript has benefited from your insightful suggestions. As accepting constructive criticisms, we have carefully addressed all the comments. Our responses to reviewers’ comments are presented in yellow highlight and given at the file.
Please see the attachment! Thanks again for your valuable comments.
- Point-by-point response to Comments
< Reviewer 1>
Comment 1: 1) Introduction - last paragraph - It is not clear what is meant by a cognitive rehabilitation that is based on self-efficacy theory and a theory can't "manage" anything. Please make this paragraph much clearer so the reader has a good idea of what is meant.
Response 1: Thank you so much for your thoughtful comments. As your comments, we edited the last paragraph of introduction section as follows:
→ Self-efficacy can be effectively manages chronic diseases by motivating health promoting behaviors [22]. Motivational outcomes of self-efficacy are choice of activities, effort, persistence and achievement [22]. Patients with stroke need to develop a sense of efficacy to complete long-term rehabilitation [23]. Self-efficacy in stroke patients helps them achieve their goals regarding successful illness management and influences their motivation for rehabilitation, thus ultimately having a positive impact on their quality of life [19, 23-24]. Higher self-efficacy can be constructed from four principal sources such as performance accomplishment, vicarious experiences, verbal persuasion, and physiological/affective states [22]. Enhancing self-efficacy of stroke patients is crucial for successful cognitive rehabilitation [23]. Therefore, this study aimed to develop a VR cognitive rehabilitation based on the self-efficacy theory and verify the effects of cognitive rehabilitation on stroke self-efficacy, cognitive function, visual perception, activities of daily living (ADL), and health-related quality of life (HRQoL).
Comment 2: 2) Methods - there needs to be a better description of the VR-based rehabilitation. There was some more description in the discussion, but it would be better put here so that the reader really understands the difference between the interventions. Also, since computer -based rehab has sometimes been considered VR, what is the difference between the computer-based cognitive rehab and the VR-based cognitive rehab.
Response 2: Thanks for your valuable comments. We agreed your comments, and thus added more detailed description of intervention including conventional, computer-assisted, virtual reality cognitive rehabilitation. Please check revised manuscript.
Comment 3: 3) Results - Table 2 is confusing - the means columns are clear, but the way the analyses columns are provided is confusing, having a time column but also a time row and then the G x T only within 1 row, etc.
Response 3: Thank you so much for your comments. We have edited the Table 2.
Comment 4: 4) Discussion - while there is a statistical difference favoring the VR group for the outcomes, often the differences between groups is quite small. This raises the issue of whether or not these are meaningful differences. Do they meet or exceed the MCID or SEM for these measures?
Response 4: Fully agree with you. Thank you so much for your thoughtful comments. For more clearly understanding comparability of between groups, we reported generalized eta square. The findings are as follows:
→ 2.6. Data Analysis
Second, repeated measures ANOVA was used to verify the effects of the study variables. Before the analysis, the assumptions of normality, homogeneity, and sphericity of the dependent variables required for repeated measures ANOVA were confirmed. The generalized eta squared (η2) in estimating effect size was used to more clearly understand comparability across between-subjects and within-subjects designs.
→ 3. Results
The results of the effects of VR cognitive rehabilitation based on the self-efficacy theory were presented in Table 2. There were significant differences among three groups in stroke self-efficacy (F=6.61, p=.003, η2=.10) and HRQoL (F=5.09, p=.009, η2=.11).
Comment 5: lines 105 and table vs line 96 - these give two different numbers for the number of enrollees. Please reconcile.
Response 5: We double-checked the number, and then edited the manuscript. Thanks for your comments.
Comment 6: line 56 - please reword "individual independent cognitive rehab. I think you mean cognitive training where individual components of cognition are trained separately?
Response 6: Accepting your comments, we edited whole sentences as follows:
→ Despite the positive outcomes of conventional training or CACR for improving cognitive function, those cognitive rehabilitation have limitations in that individual components of cognitive function are trained separately even though stroke patients necessitates a com-bination of attention, memory, and visual perception under actual daily living environ-ment [4,13-14].
Comment 7: Consort diagram - you have 2 control group I's
Response 7: Thanks for your comments. We have edited Consort diagram.
Comment 8: lines 198-199 vs 205 - did this group get 2 30-min sessions/day or was one really 15 min/day?
Response 8: Thanks for your questions.
Participants in three groups gave same amount (60 minutes) of cognitive rehabilitation. In other words, 30 minutes conventional rehab and 30 minutes (15 minutes of virtual reality cognitive rehabilitation and 15 minutes of training tasks related to VR contents using a stroke cognitive recovery workbook) of intervention was given for the experimental group. In case of control group2, they received 60 minutes of conventional rehab.
We edited some sentences to help people understand more clearly.
→ The VR cognitive rehabilitation based on the self-efficacy theory was offered five days a week for eight weeks, with a total of 40 sessions. Each session lasted 30 minutes (15 minutes of virtual reality cognitive rehabilitation and 15 minutes of training tasks related to VR contents using a stroke cognitive recovery workbook).
2.4.2. Experimental Group
The patient-specific individual program was provided five days a week for eight weeks from June to August 2022. The experimental group participated in 30 minutes of conventional cognitive rehabilitation in the morning and 30 minutes of VR cognitive re-habilitation in the afternoon.
Comment 9: line 298 - so there was a regular group discussion as part of the VR group's rehab? If so this should be in the methods. If not, then it needs to be explained how this occurred if it wasn't part of the study and could be a confound.
Response 9: Thank you so much for your comments. More detailed explanation was added into ‘Methods’ section as follows:
→ 2.4.1.
Additionally, participants in the experimental group had small group discussions of four or five persons to share their VR experiences every Friday.
2.4.2. Experimental Group
The patient-specific individual program was provided five days a week for eight weeks from June to August 2022. The experimental group participated in 30 minutes of conventional cognitive rehabilitation in the morning and 30 minutes of VR cognitive re-habilitation in the afternoon. A therapist engaged a participant one-on-one and conducted conventional cognitive rehabilitation, using papers and pencils for cognitive activities, such as cards, puzzles, calculation tasks, and picture and number matching. VR cognitive rehabilitation based on the self-efficacy theory was given to the participants in a virtual reality room. The VR cognitive rehabilitation was conducted using immersive programs with goggles and controllers, with each session lasting 30 minutes, consisting of 15 minutes of virtual reality cognitive rehabilitation and 15 minutes of individual training using a stroke cognitive recovery workbook. Furthermore, every Friday, group discussions were conducted with five participants each.
Reviewer 2 Report
Comments and Suggestions for Authors
Title: Effects of Virtual Reality Cognitive Rehabilitation for Stroke Patients: Three-arm Randomized Controlled Trial.
This three-arm randomized controlled trial aimed to compare several effects of a virtual reality cognitive rehabilitation based on a self-efficacy versus a conventional and computer-assisted cognitive therapy in stroke patients for a successful cognitive rehabilitation.
Main comments
In general, the manuscript is well-written. Some specific comments are presented below.
Specific comments
- Lines 5, 8: Replace OCRID by ORCID
0. Abstract
- Line 26: “Cognitive Rehabilitation” is not a MeSH term, instead you can use “Cognitive Training”. It would be appropriate to use 4-6 MeSH terms in the Keywords section.
1. Introduction
- Line 61: It would be interesting to add the conclusions of references about computer-assisted cognitive rehabilitation (CACR) in other cognitive disorders or diseases as Alzheimer, Parkinson, traumatic brain injury…
- Line 70: It would be interesting to add the conclusions of references about virtual reality cognitive rehabilitation based on a self-efficacy in other cognitive disorders or diseases as Alzheimer, Parkinson, traumatic brain injury…
2. Materials and Methods
- Lines 95-97: The participants data that were assessed for eligibility are not the same in lines 95-97 (n=54) and in Figure 1 (n=64). Please, check it.
- Lines 188-191: Detail between parenthesis the dynamics of the contents included in the development of self-efficacy theory based virtual reality cognitive rehabilitation. For example, in lines 320-322 “Choosing best food” is explained.
- Line 215: Explain with more details the activities carried out in the conventional cognitive rehabilitation.
3. Results
- No comments.
4. Discussion
-Lines 376-378: Delete these lines.
5. Conclusions
- Lines 381-391: Conclusions paragraph must be shorter and related to the aims in lines 70-73.
- Lines 393-401: Limitations must be a paragraph at the end of “Discussion”.
References
- Line 489: Insert DOI of reference 30.
Author Response
- Summary
We sincerely thank the reviewer's valuable comments for thorough reading of our manuscript. Our manuscript has benefited from your insightful suggestions. As accepting constructive criticisms, we have carefully addressed all the comments. Our responses to reviewers’ comments are presented in yellow highlight and given at the file.
Please see the attachment! Thanks again for your valuable comments.
- Point-by-point response to Comments
< Reviewer 2>
Comment 1: Lines 5, 8: Replace OCRID by ORCID
Response 1: Thank you so much for your comments. We corrected our error from OCRID to ORCID.
Comment 2: Line 26: “Cognitive Rehabilitation” is not a MeSH term, instead you can use “Cognitive Training”. It would be appropriate to use 4-6 MeSH terms in the Keywords section.
Response 2: We replaced “cognitive rehabilitation” with “cognitive training” with MeSH term.
Comment 3: 1. Introduction
Line 61: It would be interesting to add the conclusions of references about computer-assisted cognitive rehabilitation (CACR) in other cognitive disorders or diseases as Alzheimer, Parkinson, traumatic brain injury
Response 3: Thank you so much for your valuable comments. We added references about CACR in other cognitive disorders as follows:
→ According to systematic review and several literatures on using computer technologies for cognitive rehabilitation, it has been shown to improve cognitive function and problem-solving abilities in people with stroke, traumatic brain injury, Alzheimer's disease, and Parkinson's disease [15-18].
Comment 4: Line 70: It would be interesting to add the conclusions of references about virtual reality cognitive rehabilitation based on a self-efficacy in other cognitive disorders or diseases as Alzheimer, Parkinson, traumatic brain injury
Response 4: Thanks for your comments. Accepting your comments, we added new evidence about virtual reality as follows;
→ VR cognitive rehabilitation can effectively merge separate cognitive rehabilitation do-mains by artificially generating visual, auditory, and tactile stimuli to create complex and integrated everyday situations [19]. Furthermore, previous studies has shown promising results in using virtual reality-based cognitive rehabilitation to improve cognitive function in patients with stroke or traumatic brain injury [6,20-21].
Comment 5: 2. Methods
Lines 95-97: The participants data that were assessed for eligibility are not the same in lines 95-97 (n=54) and in Figure 1 (n=64). Please, check it.
Response 5: Thank you so much for your valuable comments. We corrected our error.
Comment 6: Lines 188-191: Detail between parenthesis the dynamics of the contents included in the development of self-efficacy theory based virtual reality cognitive rehabilitation. For example, in lines 320-322 “Choosing best food” is explained.
Response 6: Thank you so much your good comments. We added more detailed explanation as follows:
→ For the VR cognitive rehabilitation to improve the cognitive function of the experimental group, virtual reality contents (InTheTech Co., Ltd., Korea), named ‘Popping colorful balloons, Finding same fishes, Save the planet, Throwing dart and other contents,’ were modified in consideration of the level of stroke patient’s cognitive function, and ‘Stretching and aerobic exercise in the park and Choosing best foods for stroke patients’ contents were added to customized the educational needs of stroke patients. For example, the 'Finding same fishes' content involved locating and eliminating a target fish while avoiding other fish, and 'Popping colorful balloons' required removing balloons that appeared on the screen in a specific order based on their shape and color. The 'Save the planet' con-tent involved avoiding obstacles while finding and rescuing planets that matched the ones displayed on both sides. Especially in 'Stretching and aerobic exercise in the park,' participants performed several missions related to exercise while walking in the park. These missions included activities such as picking up leaves, avoiding bicycles, and giving a high five to dogs. The contents were classified into levels 1 through 5. Each of the contents was designed to training specific cognitive function such as attention, memory, visual perception, and executive function. Individualized training sessions were con-ducted, and the content varied daily to align with the cognitive domains trained in the virtual reality content. For example, activities such as rhythmic clapping, cup tapping, singing with modified lyrics, and learning rhythmic patterns were performed.
Comment 7: Line 215: Explain with more details the activities carried out in the conventional cognitive rehabilitation.
Response 7: Thank you so much for your thoughtful comments. As your comments, we added more detailed explanation as follows:
→ Five days a week, for eight weeks, control group 1 received 30 minutes of convention-al cognitive rehabilitation in the morning and 30 minutes of computer-assisted cognitive rehabilitation in the afternoon. Computer-assisted cognitive rehabilitation was administered by a therapist inside the rehabilitation therapy room by using ComCog (Neofect, Korea, 2019), where patients interacted with a computer monitor under the guidance of a therapist. The contents of ComCog consist of attention and memory training exercises, including activities like puzzle solving and piano playing, spanning 16 levels in total. ComCog were designed to be immersive and enjoyable, and provided quantitative feed-back for patients’ activities. The difficulty level of training tasks could be continually adjusted based on each patient’s performance.
Control group 2 received 30 minutes of conventional cognitive rehabilitation each morning and afternoon, five days a week, for eight weeks in the rehabilitation therapy room by an occupational therapist. The conventional cognitive rehabilitation by a therapist involved a structured approach to improve patients’ cognitive functions. Training sessions were one-one-one using paper-and pencil tasks, which consisted of puzzles, calculation, picture matching, and others.
Comment 8: (4. Discussion) Lines 376-378: Delete these lines.
Response 8: Thanks for your comments. We deleted Lines 376-378.
Comment 9: Lines 381-391: Conclusions paragraph must be shorter and related to the aims in lines 70-73.
Response 9: We have concisely summarized the conclusions, and rewrote the recommendations.
→ With a three-arm randomized controlled trial design, this study tried to build a VR cognitive rehabilitation based on the self-efficacy theory and verify its effectiveness by applying various self-efficacy strategies, such as performance accomplishments, vicarious experience, and verbal persuasion. Our findings demonstrated that the self-efficacy of stroke patients in the experimental group improved more than those in control group 1, who received conventional rehabilitation, and control group 2, who underwent computer-assisted cognitive rehabilitation. Significant differences in cognitive function, visual perception, ADL, and HRQoL were also significant.
The following recommendations are made based on the results of the current study. First, it is proposed that repeated studies be conducted to validate the effectiveness of immersive VR cognitive rehabilitation based on the self-efficacy theory for stroke patients who are admitted to various types of hospitals, such as tertiary general hospitals and general hospitals. Second, it has been reported that the cognitive function of stroke patients significantly improved up to 12 months after the stroke, but the improvement was only marginal after 36 months [31]. Therefore it is necessary to investigate the program's effect according to the onset date, such as between three and 12 months or between 12 and 36 months.
Comment 10: Lines 393-401: Limitations must be a paragraph at the end of “Discussion”.
Response 10: As your valuable comments, the sentence of limitations relocated end of “Discussion”.
→ Despite the innovative significance, our study has some limitations. First, there are limitations in generalizing the results to all stroke patients with cognitive impairment because the study was conducted on patients admitted to a rehabilitation hospital in Korea. Second, this study was conducted on patients who experienced a stroke between 3 and 36 months ago, which may limit the generalizability of the research findings.
Comment 11: (References) Line 489: Insert DOI of reference 30.
Response 11: We deleted old reference # 30, and then have replaced new reference with a better journal.
Reviewer 3 Report
Comments and Suggestions for Authors
This is an excellent study aiming to respond via a randomized study if virtual reality cognitive rehabilitation is beneficial to stroke patients. It is well written and easy to follow. I do have the following suggestions:
1. There is no need to be so specific in title regarding the methodology. Please remove “Three-arm” from the title
2. What is the rational to exclude bipolar patients from the study?
Author Response
- Summary
We sincerely thank the reviewer's valuable comments for thorough reading of our manuscript. Our manuscript has benefited from your insightful suggestions. As accepting constructive criticisms, we have carefully addressed all the comments. Our responses to reviewers’ comments are presented in yellow highlight and given at the file.
Please see the attachment! Thanks again for your valuable comments.
- Point-by-point response to Comments
< Reviewer 3>
Comment 1: There is no need to be so specific in title regarding the methodology. Please remove “Three-arm” from the title
Response 1: Thank you so much for your comments. We deleted words “Three-arm” form the title.
Comment 2: What is the rational to exclude bipolar patients from the study?
Response 2: Thanks for your valuable comments. We added a more detailed explanation and the rational for exclusion criteria as follows:
→ The specific exclusion criteria for participants were as follows: those who scored between 0 and 24 on the Korean version of the Modified Barthel Index (K-MBI); those with visuospatial neglect; those with underlying neurological conditions such as dementia; and those with a history of any psychiatric disorders due to medical concerns that immersive VR might cause exacerbate certain psychosocial symptoms by side effects such as reduced sense of presence and development of inadequate responses to the real world [25].
Round 2
Reviewer 1 Report
Comments and Suggestions for Authors
Thank you for being so attentive to the comments. The paper is much improved.
However, I have 1 small edit and 2 issues that still need attention.
1) small edit - line 69 - this line is not grammatically correct. Perhaps you mean something like "...can influence how effectively people manage chronic..."?
2) Please check that you actually used eta -squared effect sizes as SPSS give partial eta squared effect sizes.
3) While the authors attempted to address the issue of meaningful differences in gains by using an effect size statistic - that is just a statistic based on the distributions of the 2 groups and cannot speak to the clinical meaningfulness of these change difference. As the change differences were really small, it's not clear that they were clinically meaningful. Often the MCID (or at least the SEM) is used as a marker of this, for lack of a clearly validated way to determine meaning to life. As some of the changes, while statistically significant, were very small, and the group differences also very small, there needs to be some conversation in this article about whether these are actually meaningful differences or not (as statistically significant differences are not always meaningful). This is particularly important for intervention trials, where implementing interventions always cost society financially and we need to really know if they actually produce greater meaningful changes than less expensive interventions.
Author Response
- Summary
We sincerely thank for your precious time in reviewing our manuscript. Our manuscript has benefited from your insightful suggestions. The English in this manuscript has been checked and edited by a professional editor of MDPI.
As accepting constructive criticisms, we have carefully addressed all the comments. Our responses to reviewers’ comments are presented in green highlight and given at the file. Please see the attachment! Thanks again for your valuable comments.
- Point-by-point response to Comments
< Reviewer 1>
Comment 1: 1) small edit - line 69 - this line is not grammatically correct. Perhaps you mean something like "...can influence how effectively people manage chronic..."
Response 1: Thank you so much for your thoughtful comments. We edited the sentence of the line 69 as follows:
→ Self-efficacy can influence how effectively people manage chronic diseases by motivating health-promoting behaviors [22].
Comment 2: 2) Please check that you actually used eta -squared effect sizes as SPSS give partial eta squared effect sizes.
Response 2: Thanks for your comments. We edited the sentence as follows:
→ Partial eta squared (ηp2) was used to estimate the effect size to more clearly understand the comparability across between-subject and within-subject designs.
Comment 3: 3) While the authors attempted to address the issue of meaningful differences in gains by using an effect size statistic - that is just a statistic based on the distributions of the 2 groups and cannot speak to the clinical meaningfulness of these change difference. As the change differences were really small, it's not clear that they were clinically meaningful. Often the MCID (or at least the SEM) is used as a marker of this, for lack of a clearly validated way to determine meaning to life. As some of the changes, while statistically significant, were very small, and the group differences also very small, there needs to be some conversation in this article about whether these are actually meaningful differences or not (as statistically significant differences are not always meaningful). This is particularly important for intervention trials, where implementing interventions always cost society financially and we need to really know if they actually produce greater meaningful changes than less expensive interventions.
Response 3: We definitely agree with your comments, and deeply appreciate your valuable comments. Thanks to your insightful suggestions, our manuscript improved much better. Accepting your comments, we have edited discussion parts, and added a limitation about a risk of interpretation for a partial eta squared value of a small effect size. Please check the manuscript.